# Sex Biases in Cancer and Autoimmune Disease Incidence Are Strongly Positively Correlated with Mitochondrial Gene Expression across Human Tissues

**DOI:** 10.3390/cancers14235885

**Published:** 2022-11-29

**Authors:** David R. Crawford, Sanju Sinha, Nishanth Ulhas Nair, Bríd M. Ryan, Jill S. Barnholtz-Sloan, Stephen M. Mount, Ayelet Erez, Kenneth Aldape, Philip E. Castle, Padma S. Rajagopal, Chi-Ping Day, Alejandro A. Schäffer, Eytan Ruppin

**Affiliations:** 1Cancer Data Science Laboratory, Center for Cancer Research, National Cancer Institute, National Institutes of Health, Bethesda, MD 20892, USA; 2Department of Cell Biology and Molecular Genetics, University of Maryland, College Park, MD 20742, USA; 3Graduate Program in Computational Biology, Bioinformatics, and Genomics, University of Maryland, College Park, MD 20742, USA; 4Laboratory of Human Carcinogenesis, Center for Cancer Research, National Cancer Institute, National Institutes of Health, Bethesda, MD 20892, USA; 5Center for Biomedical Informatics and Information Technology, National Cancer Institute, National Institutes of Health, Bethesda, MD 20892, USA; 6Trans-Divisional Research Program, Division of Cancer Epidemiology and Genetics, National Cancer Institute, National Institutes of Health, Bethesda, MD 20892, USA; 7Department of Molecular Cell Biology, Weizmann Institute of Science, Rehovot 7610000, Israel; 8Laboratory of Pathology, Center for Cancer Research, National Cancer Institute, National Institutes of Health, Bethesda, MD 20892, USA; 9Division of Cancer Prevention, National Cancer Institute, National Institutes of Health, Bethesda, MD 20892, USA; 10Women’s Malignancies Branch, Center for Cancer Research, National Cancer Institute, National Institutes of Health, Bethesda, MD 20892, USA; 11Laboratory of Cancer Biology and Genetics, Center for Cancer Research, National Cancer Institute, National Institutes of Health, Bethesda, MD 20892, USA

**Keywords:** cancer incidence, autoimmunity, autoimmune disease incidence, inflammation, immunity, sex bias, mitochondria

## Abstract

**Simple Summary:**

Our study investigates the well-known observation/quandary that cancer occurs more frequently in men while autoimmune diseases (AIDs) occur more frequently in women. This has motivated us to explore whether these sex biases may have a common basis. To study that, we assembled and analyzed a large collection of cancer and AID incidence datasets, including matched data from 29 countries. We first, quite strikingly, find that the sex biases observed in the incidence of AIDs and cancers that occur in the same tissue are positively correlated across human tissues. To our knowledge, this is the first time this across-tissue relationship has been quantitatively demonstrated. Second, we find by analyzing healthy human tissue gene expression data that the sex bias in the expression of mitochondrial-encoded genes stands out as the common key factor whose levels across human tissues are most strongly and positively associated with both cancer and AID incidence rate sex biases, pointing to the key potential role of these genes in determining sex bias in both disorders. These findings may further prompt researchers to explore how pertaining findings in cancer studies could cross fertilize AID studies and vice versa, potentially enhancing our ability to prevent and treat these diseases.

**Abstract:**

Cancer occurs more frequently in men while autoimmune diseases (AIDs) occur more frequently in women. To explore whether these sex biases have a common basis, we collected 167 AID incidence studies from many countries for tissues that have both a cancer type and an AID that arise from that tissue. Analyzing a total of 182 country-specific, tissue-matched cancer-AID incidence rate sex bias data pairs, we find that, indeed, the sex biases observed in the incidence of AIDs and cancers that occur in the same tissue are positively correlated across human tissues. The common key factor whose levels across human tissues are most strongly associated with these incidence rate sex biases is the sex bias in the expression of the 37 genes encoded in the mitochondrial genome.

## 1. Introduction

Both autoimmune diseases (AIDs) and cancers have notably sex-biased incidence rates. Most AIDs occur more often in women [1,2], and most cancers occur more often in men [3,4,5]. While sex differences in several key biological factors have been implicated in the biased incidence rates observed for both AIDs and cancer, including inflammation and immunity, metabolism and sex hormones, their mechanistic underpinnings remain largely unexplained [2,6,7].

Given these observations, we asked whether the sex biases observed in the incidence of AIDs and cancers that occur in the same tissue are correlated *across* human tissues. This question is of fundamental interest, since an affirmative answer may suggest that there are common factors underlying their incidence. Establishing such a link between AIDs and cancers could further prompt researchers to explore how pertaining findings in AIDs could cross fertilize cancer risk studies and vice versa, potentially enhancing our ability to prevent and treat these diseases.

To explore whether these sex biases are correlated across human tissues, we collected population-based AID incidence studies for tissues that have both a cancer type and an AID that arise from that tissue. For countries for which we collected AID incidence data, we gathered incidence data for corresponding cancer types from national cancer registries. Analyzing a total of 182 country-specific, tissue-matched cancer-AID incidence rate sex bias data pairs, we find that the incidence rate sex biases observed for AIDs and cancers that occur in the same tissue are positively correlated across human tissues. In addition, we analyzed gene expression data from non-diseased tissue samples to determine if sex biases in gene set expression in these tissues are correlated with AID and cancer incidence rate sex biases in the same tissues. We find that the top positively enriched gene set across human tissues whose expression sex bias is most strongly associated with the incidence rate sex biases for AIDs, cancers, and AIDs and cancers considered jointly, is the set of 37 genes encoded in the mitochondrial genome.

## 2. Materials and Methods

### 2.1. Overview

Our analysis is divided into two main parts: curation and analysis of disease incidence rate data; and investigation of associations between incidence rates and gene expression in corresponding non-diseased human tissue samples. First, we studied the association of incidence rates for AIDs and cancers occurring in the same tissue. We collected incidence data for AIDs from published studies, and for each country for which we found incidence data for a given AID, we collected incidence data from that country’s national cancer registry for cancers occurring in the same tissue as the AID. We matched AID and cancer incidence data by tissue and by country to produce country-specific tissue-matched AID-cancer incidence rate data pairs. We then computed across-tissue correlations between AIDs and cancers for male incidence rates, female incidence rates, overall incidence rates, and incidence rate sex biases, at both the individual country level and the across-country global level.

Next, we used non-diseased human tissue transcriptomic data from GTEx version 8 [8] to investigate possible factors across human tissues that might be associated with incidence rate sex biases. We computed correlations between incidence rate sex biases and either expression of individual genes or enrichment of human functional gene sets across tissues.

### 2.2. Autoimmune Disease Incidence Data Curation

We first performed an extensive literature search for sex-specific incidence data for AIDs. For each AID, we searched for original studies mentioning the disease and epidemiology, prevalence, incidence, incidence rate, or sex bias using Google Scholar. We considered only population-based studies that use clinical inclusion criteria and have at least 25 cases for a given disease. We evaluated whether or not a study was population-based using either (a) the characteristics of the existing data source used in the study (e.g., a mandatory country-wide reporting registry) or (b) estimates showing that the data collected in the study were likely representative of the overall population. We evaluated whether or not a study used clinical diagnostic criteria by looking for use of a disease-specific blood test, a histological assay, or other evidence used to confirm diagnosis and rule out similar non-autoimmune conditions. Additionally, we considered only AIDs with a focal primary tissue (e.g., we included ulcerative colitis but excluded Crohn’s disease), for which we could find incidence data for at least three countries. We excluded sex-specific tissues.

We collected 188 AID-country incidence rate datasets from 167 studies. For each dataset, we calculated the *incidence rate sex bias* (*IRSB*) as
$$\begin{array}{c} IRSB=log_{2}\left(IR_{MALE}/IR_{FEMALE}\right)\\ \left(A\right) \end{array}$$
so that a value of zero indicates no bias, a positive value indicates a higher incidence rate in males (termed a “male bias”) and a negative value indicates a higher incidence rate in females (similarly termed a “female bias”). A majority of the studies provided sex-specific (123 of 188 datasets, 65%) and total (143 of 188, 76%) incidence rates (IR):$$\begin{array}{c} IR_{POP}=cases_{POP}/population_{POP}\\ \left(B\right) \end{array}$$
where: “POP” stands for either the “MALE”, “FEMALE”, or “TOTAL” population; $cases_{TOTAL}=cases_{MALE}+cases_{FEMALE}$; and $population_{TOTAL}=population_{MALE}+population_{FEMALE}$). Most studies reported IR as cases per year per 10^5^ persons; those using a different scale were converted to this scale. We used “crude” incidence rates (as defined above) when available; some studies provided only age-adjusted incidence rates.

#### Estimating Incidence Rates

For each of the four incidence rate measures we consider (IRSB, IR_F_, IR_M_, and R_TOTAL_) the majority of studies provided a value, while other studies gave values for other measures (i.e., different incidence rates or case counts) that can be used to estimate the value of that measure. For a given measure we can divide our AID-country datasets into four groups (Table 1): (1) those with the measure’s value but not the values of other measures we can use to estimate that value; (2) those with the measure’s value and the values of other measures we can use to estimate that value; (3) those without the measure’s value but with the values of other measures we can use to estimate that value; and (4) those with neither the measure’s value nor the values of measures we can use to estimate that value. For each measure, we assessed the accuracy of our estimator by comparing the actual and estimated values for datasets in group (2), and then used that same estimator to estimate values for datasets in group (3).

The estimators for all four measures require a value for the population’s sex ratio (Formula (1))
(1)$$\begin{array}{c} Sex_{Ratio}=\frac{population_{FEMALE}}{population_{MALE}}\\ \left(C\right) \end{array}$$

As only one study provided the background population sex ratio, we estimated measures using either a sex ratio of 1:1 or the sex ratio for the corresponding population (matching the specific country during the years the study was conducted) according to United Nations estimates [9]. Based on (A), (B), and (C), we estimated *IRSB*, *IR_F_*, *IR_M_*, and *IR_TOTAL_* as (Formulas (2)−(5)):(2)$$\begin{array}{c} IRSB=log_{2}\left(\frac{cases_{MALE}}{cases_{FEMALE}}\times Sex_{Ratio}\right) \end{array}$$
(3)$$IR_{FEMALE}=IR_{TOTAL}\times \frac{cases_{FEMALE}}{cases_{TOTAL}}\times \left(1+1/Sex_{Ratio}\right)$$
(4)$$IR_{MALE}=IR_{TOTAL}\times \frac{cases_{MALE}}{cases_{TOTAL}}\times \left(1+Sex_{Ratio}\right)$$
(5)$$IR_{TOTAL}=IR_{M}\times \left(\frac{1}{1+Sex_{Ratio}}\right)+IR_{F}\times \left(\frac{Sex_{Ratio}}{1+Sex_{Ratio}}\right)$$

To assess the accuracy of our estimators we compared the actual and estimated values for datasets in group (2) in two ways (Table 2). First, we computed the Pearson’s correlation coefficient *r* between the two values. All estimators were accurate: for each the correlation coefficient was close to 1 and the one-sided *t*-test was significant. Second, we computed a simple linear model of the form $x_{actual}=\beta \times x_{estimate}+\alpha$. All estimators were accurate: for each the coefficient β was close to 1 and the *r^2^* close to 1 (where *r* is the Pearson’s correlation coefficient). For all four measures the estimators performed well, but for each measure the estimator using a sex ratio of 1:1 performed as good as or slightly better than the estimator using the sex ratio based on the United Nations estimates. Accordingly, for our analyses we used estimators with a sex ratio of 1:1. For all of our analyses, results computed using only given values, and not estimates, were consistent with results computed using both given and estimated values (the code for this paper includes scripts to reproduce all tests and figures using data that either includes or excludes estimated values).

When multiple studies were available for an AID in a country, we used the across-study arithmetic mean of each incidence rate measure as the measure value for that AID-country pair (for $IR_{MALE}$, $IR_{FEMALE}$, $IR_{TOTAL}$, or IRSB measures). Overall, surveying 167 published studies (Supplementary References), we calculated 133 country-specific AID incidence rate sex bias data points for 17 AIDs in 33 countries (Appendix A, e.g., the mean incidence rate sex bias for Type 1 diabetes in Spain is one such data point).

### 2.3. Cancer Incidence Data Curation

Cancer incidence rates were calculated from GLOBOCAN [10] data for all but three countries. For each country for which we had AID data, we computed each cancer type’s incidence rate measure for each year and then averaged the yearly measure values to produce a single measure value for each country-cancer pair (for $IR_{MALE}$, $IR_{FEMALE}$, $IR_{TOTAL}$, or IRSB measures). Cancer data for Finland [11], Sweden [12], and Taiwan [13] were collected from country-specific databases. For Finland and Sweden we calculated each incidence rate measure as the across-year average yearly measure for each cancer type for the most recent 20 years (1999–2019) for each country. For Taiwan we calculated each measure as the average of the measure for the two available time periods (1998–2002, 2003–2007). Overall, we calculated 165 country-specific cancer incidence rate sex bias data points for 17 cancer subtypes in 29 countries (Appendix A; for an additional four countries we were unable to find population-level cancer incidence data).

### 2.4. Pairing AID and Cancer Incidence Data

Across 12 human tissues we paired 17 AIDs with 17 cancer types for a total of 24 cancer-AID data pairs. To compute the correlation between AIDs and cancer incidence rate sex biases across tissues, we grouped AIDs with matched cancers occurring in the same tissue in the same country (Appendix A). For example, for the UK, we paired thyroid AID data points for Hashimoto’s hypothyroidism and Graves’ hyperthyroidism with cancer data points for thyroid carcinoma and thyroid sarcoma, resulting in 4 possible thyroid cancer-AID pairs. The 133 country-specific AID incidence rate sex bias data points were matched to the 165 country-specific cancer incidence rate sex bias data points, yielding a total of 182 country-specific, tissue-matched cancer-AID incidence rate sex bias data pairs that are jointly present in both the AID and cancer datasets (Appendix A).

### 2.5. Gene Expression Analysis of Human Tissues

Gene expression was calculated from GTEx v8 data [8] provided in transcripts-per-million (TPM). For gene *i* and tissue *k* with *m* samples we calculated the within-tissue gene expression (GE) as the arithmetic mean TPM across samples as (where “POP” stands for either the “MALE”, “FEMALE”, or “TOTAL” population, Formulas (6) and (7)):(6)$$GE_{POP,i,k}=\frac{1}{m}\overset{m}{\underset{j=1}{\sum }}TPM_{POP,i,j,k}$$
and the gene expression sex bias (ESB) as:(7)$$ESB_{i,k}=log_{2}\left(GE_{MALE,i,k}/GE_{FEMALE,i,k}\right)$$
where both $GE_{MALE,i,k}$ and $GE_{FEMALE,i,k}$ are positive.

For a set of *n* genes *N* and tissue *k* with *m* samples, we calculated the within-tissue gene set activity (GSA) as the geometric mean of gene expression across genes (Formulas (8) and (9)):(8)$$GSA_{POP,N,k}={\left(\overset{n}{\underset{i=1}{\prod }}GE_{POP,i,k}\right)}^{1/n}$$
and the gene set activity sex bias (ASB) as:(9)$$ASB_{N,k}=log_{2}\left(GSA_{MALE,N,k}/GSA_{FEMALE,N,k}\right)$$
where both $GSA_{MALE,N,k}$ and $GSA_{FEMALE,N,k}$ are positive.

### 2.6. Gene Set Enrichment Analysis across Human Functional Pathways

We performed gene set enrichment analysis (GSEA) in three steps. First, for each gene in each tissue we calculated the gene expression sex bias (ESB). Second, we computed the across-tissue Spearman correlation of the ESB of each gene with AID or cancer IRSBs (we abbreviate these correlations as $corr_{ESB/IRSB}$). We also computed aggregated or “joint” $corr_{ESB/IRSB}$ values as the average for each gene of its $corr_{ESB/IRSB}$ value for AID IRSBs and its $corr_{ESB/IRSB}$ value for cancer IRSBs. Finally, for each of these three phenotypes, we ordered all the genes from greatest to least by the $corr_{ESB/IRSB}$ values and performed a GSEA [14] to identify gene sets and pathways that were either significantly positively or negatively associated with IRSB (for gene sets used see Results). We considered GSEA results significant if the adjusted $p\leq 10^{-3}$ (we used the Benjamini-Hochberg method to adjust *p*-values for multiple tests) and ranked the results by normalized enrichment score (NES) [14].

## 3. Results

We surveyed 167 published AID studies and the cancer registries for 29 countries to assemble 182 country-specific, tissue-matched cancer-AID incidence rate sex bias data pairs (Methods; Appendix A). For each study, we calculated the *incidence rate sex bias* (*IRSB*) as $IRSB=log_{2}\left(IR_{MALE}/IR_{FEMALE}\right)$, where $IR_{MALE}$ and $IR_{FEMALE}$ are the male and female incidence rates, so that a value of zero indicates no bias, a positive value indicates a higher incidence rate in males (termed a “male bias”) and a negative value indicates a higher incidence rate in females (similarly termed a “female bias”). Having assembled these data, we computed the mean IRSBs to get a view of tissue-matched cancer and AID incidence rate sex bias across tissues, yielding global IRSB values for 17 AIDs and 17 cancer types across 12 human tissues, comprising a total of 24 cancer-AID data pairs. As expected, most AID incidence rates are female-biased (a negative sex-bias score), while most cancer incidence rates are male-biased (a positive sex-bias score) (Figure 1A, Appendix A). Figure 1B presents the correlation of the IRSB of these disorders across human tissues, summed up across all countries surveyed. Notably, we find an overall positive correlation (Pearson correlation *r* = 0.48 with two-sided *t*-test *p* = 0.017, Spearman correlation *r* = 0.43 with two-sided *t*-test, *p* = 0.034). Repeating this analysis using various levels of cancer type classification shows a consistent and robust correlation (Appendix A, Appendix A). (We used Pearson’s product-moment correlation coefficient to measure correlation because it takes effect size into account. We also provide correlation test results based on Spearman’s rank correlation coefficient as this assesses correlation differently and may be of interest to the reader. We considered a correlation test result significant when the *t*-test adjusted p≤0.05. We used the Benjamini-Hochberg method to adjust *p*-values for multiple tests). Second, studying this correlation in a country-specific manner for the four countries with at least 18 AID-cancer data pairs, we find a country-specific significant correlation for Sweden, while the correlations for Denmark, the UK and the USA have *q*-values (*p*-values corrected for multiple hypotheses testing) > 0.05 but are quite close to this threshold, showing a consistent trend for each country (Figure 1C).

Observing this fundamental correlation, we next asked if we could identify factors that might jointly modulate both the incidence rate sex bias observed in cancer and in AID across human tissues. We conducted both an unbiased general investigation and a hypothesis-driven one. We specifically examined four major factors that have been previously associated in the literature with the incidence rates of cancers and AIDs and/or their incidence rate sex biases. Those include (1) inflammatory or immune activity in the tissue [15,16]; (2) expression of immune checkpoint genes [17,18]; (3) the extent of X-chromosome inactivation [6,19]; and finally, (4) mitochondrial activity [20,21] and mitochondrial DNA copy number [22,23].

Having these literature-driven specific hypotheses in mind, we still have chosen to begin by systematically charting the landscape of gene sets whose sex-biased enrichment in normal tissues is associated with IRSB in cancers and AIDs in an unbiased manner (see Section 2.5). We analyzed gene expression data from non-diseased tissue samples from GTEx v8 [8], for tissues in which both cancer and AID arise; GTEx data were available for 10 of the 12 tissues we studied above (Appendix A). First, (1) for each gene in each tissue we calculated the *expression sex bias (ESB)* as $ESB=log_{2}\left(GE_{MALE}/GE_{FEMALE}\right)$, where $GE_{MALE}$ or $GE_{FEMALE}$ denote the average gene expression in TPM (transcripts-per-million) for male or female samples of the tissue. (2) Second, we computed the correlation of the expression sex bias of each gene with AID or cancer IRSBs (we abbreviate these correlations as $corr_{ESB/IRSB}$). We also computed aggregated or “joint” $corr_{ESB/IRSB}$ values as the average for each gene of its $corr_{ESB/IRSB}$ value for AID IRSBs and its $corr_{ESB/IRSB}$ value for cancer IRSBs. (3) Finally, for each of these three phenotypes, we ranked all the genes from top to bottom by the *corr_ESB/IRSB_* values and performed a gene set enrichment analysis (GSEA) [14,24] to identify gene sets and pathways that were either significantly positively or negatively associated with IRSB. In total, this analysis covered 7763 gene sets, including gene ontology biological process sets and chromosome-location based sets from MSigDB [25], three X-chromosome gene sets (fully escape X-inactivation, variably escape X-inactivation, and pseudoautosomal region) [26], and finally, the two separate sets of nuclear-encoded genes whose protein products localize to the mitochondria and the 37 mitochondrial-genome-encoded genes [27]. In this study, we consider the 37 mitochondrial genes as a set; all the genes are listed individually and some previously identified associations between individual genes and either AIDs or cancer are shown in Appendix A.

Figure 2 shows the top positively and negatively $corr_{ESB/IRSB}$ enriched sets with *p* ≤ 10^−3^ after multiple hypotheses test correction for AID incidence (positive, Panel A; negative Panel B), for cancer incidence (positive, Panel C; negative Panel D), and their joint aggregate enrichment for both AID and cancer incidence (positive, Panel E, negative, Panel F). Strikingly, the top enriched gene set (highest normalized enrichment score (NES)) in *all* three phenotypes is the set of 37 genes encoded on the mitochondrial genome, including many genes with high $corr_{ESB/IRSB}$ values. In contrast, while the (much larger) set of all genes encoding proteins that localize to the mitochondria is significantly enriched for cancer IRSB, it is not significantly enriched for AID IRSB, where it is only ranked 3842 out of 6420 (negatively) enriched gene sets. Several immune-related gene sets also show high and significant $corr_{ESB/IRSB}$ positive enrichments in accordance with one of our initial hypotheses (Figure 2). However, the three different X chromosome gene sets studied in light of another one of our original hypotheses are not significantly enriched in $corr_{ESB/IRSB}$ values. Finally, several mRNA processing gene sets show strong negative significant correlations and high negative NES scores with AID and cancer incidence.

To obtain a clearer visualization of the key positively enriched gene sets described above, we summarized the expression of the genes composing a given gene set in a normal GTEx tissue by computing their geometric mean, giving us a single activity summary value (see Section 2.5). We then computed the correlation across tissues between these summary values of the gene sets in each normal tissue and the IRSBs of cancers or AIDs (Figure 3). In concordance with the results of the unbiased analysis presented above, we do not observe a significant correlation between cancer or AID incidence rate sex bias and the expression of key immune checkpoint genes (CTLA-4, PD-1, or PD-L1, Appendix A), or the extent of X-chromosome inactivation (quantified by the expression of XIST lncRNA [28], Appendix A). We also do not find such significant consistent correlations for the top immune gene sets found via the unbiased analysis (previously shown in Figure 2). However, we do find strong correlations between these summary values for the mitochondrial gene set, which was ranked highest in Figure 2 (gene set “MT”): Remarkably, we find that the sex bias of mtRNA expression in GTEx tissues is positively correlated both with AID incidence rate sex bias (Pearson *r* = 0.56, one-sided *t*-test *p* = 0.018) and with cancer incidence rate sex bias (Pearson r = 0.67, one-sided t-test *p* = 0.0058) (Figure 3A,B; the correlations between mtRNA expression and cancer and AID incidence rates for each of the sexes individually are provided in Appendix A). The significance of these two associations is further supported by observing that the basic correlation between cancer and AID IRSBs becomes insignificant when we compute the partial correlation between these two variables while controlling for the mtRNA expression bias (Pearson r = 0.21, two-sided *t*-test *p* = 0.42). Overall, these findings are in line with previous reports linking mitochondrial activity [20,21] and mtDNA copy number [22,23] with higher AID and cancer incidence.

## 4. Discussion

The correlative findings between the expression of mitochondrially encoded genes and cancer and AID IRSBs across human tissues are quite surprising, giving rise to two further fundamental questions. First, what biological mechanisms may be associated with sex differences in overall mitochondrial functioning? One potential candidate may be estrogen signaling, which has been shown to regulate at least four mitochondrial functions relevant to health and disease [29], including, (1) biogenesis of mitochondria, whose levels differ across sexes and tissues [30], (2) T-cell metabolism (including mitochondrial activity measured by Seahorse assays) and T-cell survival (estimated by retention of inner membrane potential) [31], (3) unfolded protein response [32] (mediated partly via mitochondrial superoxide dismutase) [33], and (4) generation of reactive oxygen species (ROS) [34]. Second, how might sex differences in mitochondria functioning modulate the sex-biased incidence observed in cancers and AIDs? One possible mechanism is through differences in ROS production, which notably involves quite a few mitochondrially encoded genes: Increased mitochondrial ROS generation has been associated with both the initiation and intensification of autoimmunity in several organ-specific AIDs [20] and with cancer initiation and progression [21]. More generally, alterations in mtDNA copy number have been associated with increased risk of lymphoma and breast cancer, [22] and somatic mtDNA mutations producing mutated peptides may trigger autoimmunity [23].

Our analyses have a few limitations and we list three main ones. First, the majority of our AID-cancer data pairs are from European countries (113 of 182 [62%]), which might introduce geographic, ethnic, or social biases. This geographic bias in our AID-cancer data pairs is largely due to a paucity of suitable AID epidemiological studies based on populations outside of North America and Europe, particularly on populations in low- and middle-income countries [35,36]. For example, despite the geographically widespread study of common AIDs such as multiple sclerosis and type 1 diabetes, for other common AIDs such as Hashimoto’s hypothyroidism, Graves’ hyperthyroidism, and ulcerative colitis, data from regions outside North America and Europe are sparse [37]. Second, factors beyond biological drivers, such as sex differences in the propensity to seek medical care or reporting of specific diseases, are not characterized in the datasets studied. However, putative disease-specific effects may be somewhat mitigated given the opposite tendency of sex biases for AIDs and cancers in a study of tissue-specific correlations like ours. Although there is evidence for the role of environmental exposures in the development of some AIDs [38], there is little evidence for sex-specific exposures contributing to sex biases in AID incidence [39]. Likewise, a recent study of the contribution of risk factors to sex disparities in the incidence of solid tumor cancers at 21 anatomical sites found that differences between male and female incidence rates are largely unexplained by factors outside of sex-related biological factors [40]. Third, although much of the incidence rate data is age-standardized, we could not take additional steps to account for age-related incidence rate differences as the sample sizes available are too small to enable doing such an analysis in a robust manner.

It has been hypothesized that chronic inflammation leads to cancer [41,42]. Indeed, a recent study analyzing UK Biobank data found positive associations between several tissue-specific immune-mediated diseases (i.e., diseases such as asthma and myositis in addition to AIDs) and subsequent cancer risk in the same individual [43]; they did not however consider sex-bias. Both this study and our study seek statistical evidence of shared risks between immune diseases and cancer. However, in contrast to this individual-level study design assessing within-subject risk of sequential disease appearance in the UK population, we chose a population-level study design assessing within-population correlations between IRSBs for AIDs and cancers affecting the same tissues across populations. A strength of our study is that when AID studies reported whether individuals developing AIDs had previous immune-related diseases or cancers we excluded individuals with such previous diseases from our study.

As in humans, sex differences have been reported in animal studies of diseases, which has prompted us to search the literature and survey previous studies of sex bias in disease incidence in rodent models of cancers and AIDs. We focused on studies of sex difference in spontaneous and/or autochthonous carcinogenesis by either carcinogen treatment or genetic engineering, excluding transplantation of syngeneic animals because these animals do not model disease development (representative examples are listed in Appendix A for cancer and AIDs, respectively). Appendix A lists our cancer incidence findings, where the sex bias was male skewed in colon, liver, kidney, pancreas, and stomach, and higher in females in the thyroid, consistent with the human reports. Interestingly, for colon, liver, kidney, pancreas, and thyroid, the sex bias disappeared or was reduced when the animals were subjected to castration/ovariectomy or hormone treatment, supporting the notion that the differences in these organs are likely to be driven by sex hormones. Appendix A lists AID rodent models that allow for direct comparisons to the human data. The AID sex bias reported is however generally higher in males than in females, in difference from the human findings, but the higher male bias observed in kidney, colon, pancreas, and skin compared to the thyroid is maintained.

## 5. Conclusions

In summary, we find a surprising overall positive correlation between cancer and AID incidence rate sex biases across many different human tissues. Among key factors that have been previously associated with sex bias in either AID or cancer incidence, we find that the sex bias in the expression of mitochondrially encoded genes (and possibly in the expression of a few immune pathways) stands out as a key factor whose aggregate level across human tissues is quite strongly associated with these incidence rate sex biases. Our findings thus call for further mechanistic studies on the role of mitochondrial gene expression in determining cancer and AID incidence and their incidence rate sex biases.

## Figures and Tables

**Figure 1 cancers-14-05885-f001:**
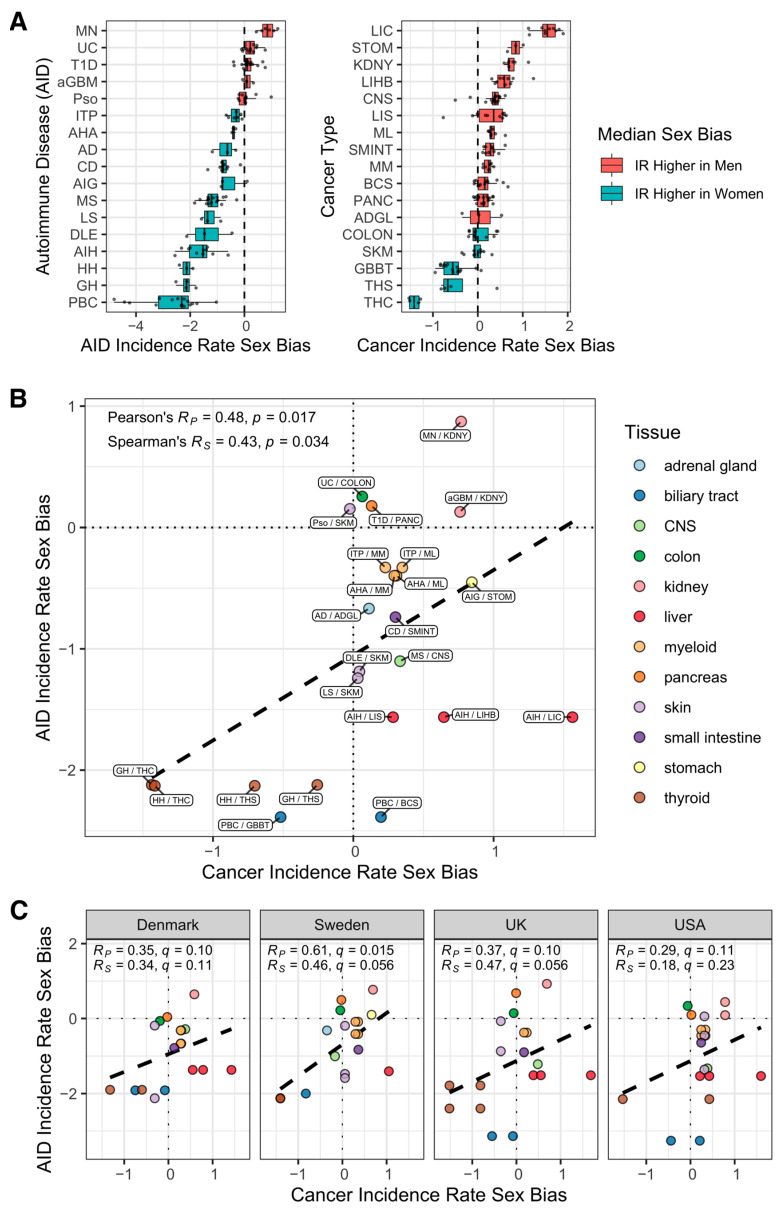
Incidence rate sex biases for cancers and AIDs are positively correlated across tissues of origin. (**A**) Distribution across countries of incidence rate sex bias (*x*-axis) for 17 AIDs and 17 cancer types (*y*-axis). All data points are shown. Box shows interquartile range (IQR, first quartile to third quartile), with center bar representing the median (second quartile). Lefthand whisker extends from first quartile (Q1) to Q1−1.5×IQR or to the lowest value point, whichever is greater. Righthand whisker extends from third quartile (Q3) to Q3 + 1.5×IQR or to the highest value point, whichever is smaller. Positive median sex bias (red) indicates median with higher incidence rate in men; negative median sex bias (blue) indicates median with higher incidence rate in women. To fit the data compactly, the x-axes for the left and right panels differ in two unconventional ways: First, the center point for (**A**) is close to −2 because most AID IRSB values are negative whereas the center point for (**B**) is just above 0 because most cancer IRSB values are positive; Second, the range of the x-axis is greater in (**A**) than it is in (**B**) (and thus the numbering is also different) because the range of AID IRSB values is greater than that of the cancer IRSB values. (**B**,**C**) Tissue-matched incidence rate sex biases for cancers (*x*-axis) and for autoimmune diseases (*y*-axis) are displayed across different tissues of origin (circle color indicates the tissue). Positive values in each of the axes indicate male bias; negative values indicate female bias. The dashed line is the simple linear regression line. Statistics in the top left corner include the Pearson’s product-moment correlation *r*-value (R_p_) and *t*-test *p*-value; and the Spearman’s rank correlation coefficient value (R_s_) and a *t*-test *p*-value (*t*-tests were two-sided for the global-level tests and one-sided for the country-level tests). For country-level tests, *p*-values were corrected for multiple testing using the Benjamini-Hochberg method to produce q-values. (**B**) Across-population averages, with the cancer-AID pairs labeled. To fit the data compactly, the plot is centered close to (0, −1) as opposed to the more conventional center of (0,0) because the majority of the AID IRSB values are negative. (**C**) Population-level data for the four countries with the largest numbers of data pairs (at least 18 out of 24 cancer-AID pairs), maintaining the tissue color labels used in the top panel (USA, 20 pairs; Denmark, Sweden, & UK, each 18 pairs). **AIDs**: AD, Addison’s disease; aGBM, anti-glomerular basement membrane nephritis; AHA, Autoimmune hemolytic anemia; AIG, Autoimmune gastritis; AIH, Autoimmune hepatitis; CD, Celiac disease; DLE, Discoid lupus erythematosus; GH, Graves’ hyperthyroidism; HH, Hashimoto’s hypothyroidism; ITP, Immune thrombocytopenic purpura; LS, Localized scleroderma; MN, primary autoimmune membranous nephritis; MS, Multiple sclerosis; PBC, Primary biliary cholangitis; Pso, Psoriasis; T1D, Type 1 diabetes; UC, Ulcerative colitis. **Cancers**: ADGL, adrenal gland cancer; BCS, liver (biliary) cholangiosarcoma; COLON, colon cancer; CNS, central nervous system cancer; GBBT, gallbladder & biliary tract cancer; KDNY, kidney cancer; LIC, liver carcinoma; LIHB, liver hepatoblastoma; LIS, liver sarcoma; ML, myeloid leukemia (acute and chronic); MM, multiple myeloma; PANC, pancreatic cancer; SKM, skin melanoma; SMINT, small intestine cancer; STOM, stomach cancer; THC, thyroid carcinoma; THS, thyroid sarcoma.

**Figure 2 cancers-14-05885-f002:**
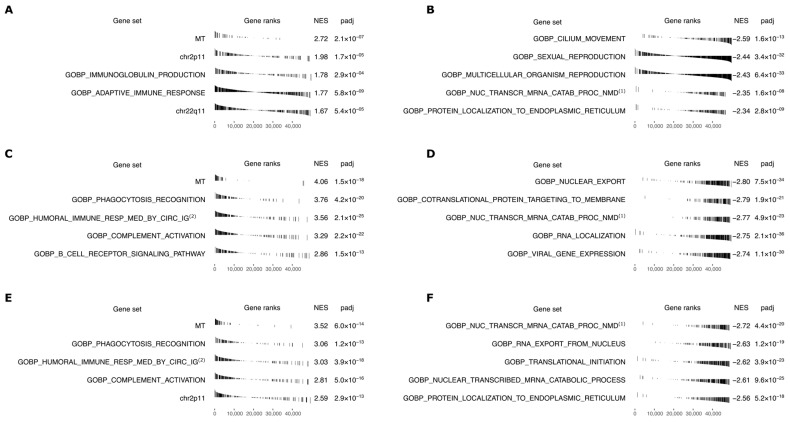
GSEA results for correlations of gene expression sex bias with IRSB across 10 GTEx tissues (adrenal gland, brain, colon, kidney, liver, pancreas, skin, small intestine, stomach, and thyroid). Top 5 positively and negatively enriched gene sets, with adjusted *p* ≤ 10^−3^, for AID incidence (positive, (**A**); negative, (**B**)), cancer incidence (positive, (**C**); negative, (**D**), and AIDs and cancers jointly (positive, (**E**); negative, (**F**)). For each gene set the plot shows: (Gene set) name; (Gene ranks) bar plot of corr_ESB/IRSB_ values ordered from highest correlation at the left to lowest at the right (bars for genes in the gene set are black); (NES) normalized enrichment score; and (padj) Benjamini-Hochberg corrected *p*-value. Abbreviated gene set names: (1) Gene Ontology Biological Process (GOBP) Nuclear transcribed mRNA catabolic process nonsense-mediated decay; (2) GOBP Humoral immune response mediated by circulating immunoglobulin; (3) GOBP Nuclear transcribed mRNA catabolic process.

**Figure 3 cancers-14-05885-f003:**
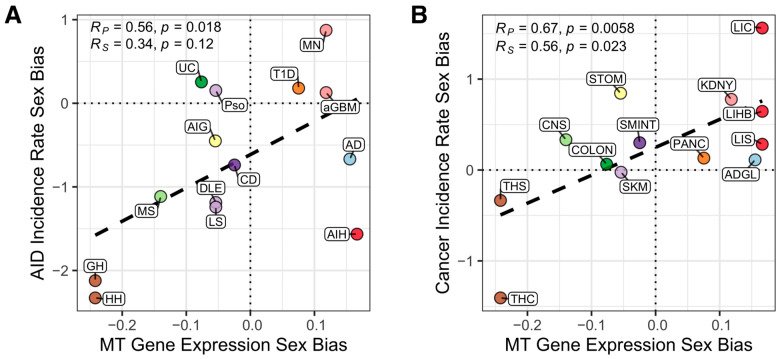
Mitochondrial gene expression is a strong correlate of sex biases in incidence rate of autoimmune diseases and cancer types across tissues. The correlation between expression ratio of mitochondrial gene expression in male vs. female tissues (*x*-axis) with the incidence rate sex biases of (**A**) autoimmune diseases (*y*-axis) and (**B**) cancer types (*y*-axis) across human tissues (circle color indicates the tissue). AIDs: AD, Addison’s disease; aGBM, anti-glomerular basement membrane nephritis; AIG, Autoimmune gastritis; AIH, Autoimmune hepatitis; CD, Celiac disease; DLE, Discoid lupus erythematosus; GH, Graves’ hyperthyroidism; HH, Hashimoto’s hypothyroidism; LS, Localized scleroderma; MN, primary autoimmune membranous nephritis; MS, Multiple sclerosis; Pso, Psoriasis; T1D, Type 1 diabetes; UC, Ulcerative colitis. Cancers: ADGL, adrenal gland cancer; CNS, central nervous system cancer; COLON, colon cancer; KDNY, kidney cancer; LIC, liver carcinoma; LIHB, liver hepatoblastoma; LIS, liver sarcoma; PANC, pancreatic cancer; SMINT, small intestine cancer; SKM, skin melanoma; STOM, stomach cancer; THC, thyroid carcinoma; THS, thyroid sarcoma.

**Table 1 cancers-14-05885-t001:** Measures to estimate, measures needed for estimators, and numbers of datasets with values for these measures. Numbers indicate dataset count and percentage (out of 188 total datasets) for each group of datasets (1–4) described in the text.

(a) Measure to Estimate	(b) Measures Needed for Estimator	(1) Datasets with (a)	(2) Datasets with (a) & (b)	(3) Datasets with (b) but Not (a)	(4) Datasets with Neither (a) nor (a)
*IRSB*	*cases_M_/cases_F_*	125 (66%)	105 (56%)	63 (34%)	0 (0%)
*IR_M_, IR_F_*	*IR_TOTAL_, cases_M_/cases_F_*	123 (65%)	84 (45%)	41 (22%)	24 (13%)
*IR_TOTAL_*	*IR_M_, IR_F_*	143 (76%)	101 (54%)	22 (12%)	23 (12%)

**Table 2 cancers-14-05885-t002:** Pearson’s correlation coefficient and simple linear model results for estimators. For each measure, each of two estimators (with sex ratio as 1:1 or from the United Nations estimates) is shown with its simple linear model coefficient *β*, intercept *α*, and r^2^, and its Pearson’s correlation coefficient r and one-sided *t*-test *p*-value.

Measure	Estimator	*β*	*α*	*r* ^2^	*r*	*p*
IRSB	$IRSB_{1:1}$	0.922	−0.0254	0.966	0.983	6.04 × 10−78
	$IRSB_{UN}$	0.923	−0.0585	0.963	0.981	5.54 × 10−76
$IR_{FEMALE}$	$IR_{FEMALE,1:1}$	1.020	−0.303	0.997	0.998	1.15 × 10−107
	$IR_{FEMALE,UN}$	1.035	−0.307	0.997	0.999	5.04 ×10−105
$IR_{MALE}$	$IR_{MALE,1:1}$	0.975	0.228	0.997	0.999	6.10 × 10−108
	$IR_{MALE,UN}$	0.959	0.236	0.997	0.999	2.26 × 10−105
$IR_{TOTAL}$	$IR_{TOTAL;1:1}$	1.013	−0.123	1.000	1.000	1.09 × 10−182
	$IR_{TOTAL;UN}$	1.013	−0.136	1.000	1.000	1.39 × 10−182

Combining data for each country.

## Data Availability

All data used is publicly available. Code and data (including links to original sources, raw data downloaded from those sources, and processed data files) used for analysis are available on Zenodo at DOI: https://doi.org/10.5281/zenodo.7058954 (last accessed on 16 September 2022). Statistical analyses and figure preparation were performed on a Macintosh computer (OS 12.5.1; 32 GB memory; 8-core 2.3 GHz processor) in RStudio (v2021.09.0 + 351 “Ghost Orchid” release) [44] running the R language (v4.1.2) [45] (R package versions used are listed in the Zenodo repository). Plots produced in R were aligned and lettered using Inkscape (v1.0) (inkscape.org (last accessed on 16 September 2022)) to produce multi-plot figures.

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
