# Peer review of "Sex Biases in Cancer and Autoimmune Disease Incidence Are Strongly Positively Correlated with Mitochondrial Gene Expression across Human Tissues"

_cancers, 2022, doi:10.3390/cancers14235885_

Round 1
Reviewer 1 Report
Autoimmune diseases (AIDs) and cancers have notably sex-biased incidence rates. Is there an association between Gender's pair AID and cancer? It's unclear, currently. In this manuscript, David R. Crawford and colleagues analyzed ~200 country-specific, tissue-matched cancer-AID incidence rate sex bias data pairs and calculated the IRSB to find most AID incidence rates are female-biased while cancer rates are male-biased. They highlight the overall positive correlation between cancer AID IRSB and MT genes can be a key factor in these incidence rate sex biases. Overall, this study helps the researchers in AID and cancer field to understanding the sex bias in these two disease types. However, the authors further need to address the following concerns.
1. It’s confusion for the reviewer, the figure 1A showed most AID incidence rates are female-biased while cancer rates are male-biased. Did this suggest the sex-bias score of AID is negatively correlated with the sex-bias score of cancer? Why is the IRSB correlation between AID and cancer still positive in Figure 1B?
2. The authors clarified that mt activity is associated with higher AID and cancer risk (lines 367-368). In their study, they calculated the incidence rate of sex bias but not AID or cancer risk.
3. Could the IRSB be affected by other confounding factors?
4. The GSEA results for the correlation of sex bias in gene expression with IRSB in Figure 2 are the combined samples of all tissues or which tissues in GTEx? Are there consistent results across different tissues?
Reviewer 2 Report
The topic is interesting and the bioinformatics and statistical analyses are well done. However, without an experimental part, the paper seems incomplete.
In particular, in my opinion, it would be important to revise the following points to try to increase the significance of the content and the scientific soundness:
- As also stated by the authors, the paper has important limitations: it was not possible to do age stratification since the available dataset was not very large; moreover, the dataset comes mostly from Europe, so the results cannot be said to be globally findable. By approaching the work only in bioinformatics terms, the starting dataset is not very meaningful: is it not possible to expand it?
- The work would have a higher significance and resonance if the bioinformatic data could also be confirmed experimentally. Specifically, since the authors found that "the sex bias in the expression of mitochondrially encoded genes (and possibly in the expression of a few immune pathways) stands out as a key factor whose aggregate level across human tissues is quite strongly associated with these incidence rate sex biases", it would be much more comprehensive study possible variations in any of these immune pathways to confirm the bioinformatic data.
Reviewer 3 Report
In this manuscript, Crawford et al analyzed the publicly available data set to draw a correlation between AID and cancer based on sex bias and what are the common factors regulating this phenomenon. By integrating the matched tissue samples of both AID and cancer incidence. They found that there is a sex bias of AID towards female and cancer towards male. Further, they have identified 37 mitochondrial genes which are associated with this sex bias. However, I have the following comments.
1- The authors should specifically mention percent incidence of both AID and cancer occurrence in male vs female which will strengthen their claim as well as will be easy for the readers to follow.
2- Please describe the characteristics of tissue samples which were selected for both AID and cancer. For example, colon tissue were selected for both colon cancer and UC. What is the incidence of UC/colon cancer in male vs female?
3- It has been hypothesized that chronic inflammation leads to cancer. UC/CD are associated with colon cancer incidences. How will the sex bias hypothesis fit into this scenario?
4- The author should give a table of the 37 mitochondrial genes and its relation to AID/cancer.
Round 2
Reviewer 3 Report
No more comments